# The Relationship between Demographics and Knowledge Risk Perception of High School Teachers: Training as a Mediator

**Michele Borgia [1], Eugenia Nissi [2], Maura La Torre [1,\*] and Guido Ortolani [3]**

[1] Department of Management and Business Administration, University "G. D'Annunzio" of Chieti-Pescara, 65127 Pescara, Italy
[2] Department of Economics, University "G. D'Annunzio" of Chieti-Pescara, 65127 Pescara, Italy
[3] State Higher Education Institute "G. Peano-C. Rosa Nereto", 64015 Nereto, Italy
[\*] Correspondence: maura.latorre@unich.it

**Abstract:** As a knowledge-based career, teachers can be exposed to knowledge risks. Since risk perception is the product of the experiences, values, memories and ideologies of individuals, the ways of perceiving knowledge risks could be useful for setting up prevention and mitigation strategies for these kinds of risks. The present paper aimed at analyzing the relationship between the demographics and the knowledge risk perception of high school teachers. The role of a teacher's training as a mediator of said relationship was analyzed as well. Using a sample of high school teachers working in Italian schools, a questionnaire was administered to gather data, and structural equation modeling analysis was employed to test the hypotheses. The results showed that demographics had a significant effect on teachers' knowledge risk perception and that training mediated this relationship. The study could be helpful for educational institutions that want to train their teachers to be prepared to face risky events related to knowledge management.

**Keywords:** demographics; knowledge risks; risk perception; training; high school; teachers; structural equation modeling

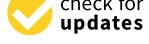



## 1. Introduction

Knowledge risks can arise from improper knowledge management that brings out the risky side of knowledge (La Torre 2020b; Seidl 2007). They can concern knowledge loss (Daghfous et al. 2013; Massingham 2018), knowledge waste (Ferenhof et al. 2015), knowledge hoarding (Dash et al. 2022; Oliveira et al. 2021), knowledge unlearning (Durst and Zieba 2017), knowledge outsourcing (Durst and Zieba 2017), knowledge forgetting (Vasileiou and Yeoh 2022; Zhang and Zhou 2009) and knowledge digitization (Durst and Zieba 2017; Ivanova et al. 2019). All of these types of knowledge risk may be harmful to organizations of any type and size, as they can occur in daily operations as well as in extraordinary ones, affecting any function or part of the organizational structure (Zieba et al. 2021b). Several actions can be taken to prevent and mitigate knowledge risks, such as the implementation of knowledge management systems and the promotion of behaviors aimed at knowledge conservation, storage, protection and sharing (Durst and Zieba 2020). Risk perception, referring to the possibility of recognizing opportunities and threats, can represent the first step in the prevention of various types of risk, including knowledge risks. Risk perception may be affected by several factors: age, gender, social status, experience and education may influence how individuals perceive risks (Brown et al. 2021; Kim et al. 2018; Rattay et al. 2021). Risk perception is also considered a social and cultural construct going beyond the individual as an expression of the values, history and ideologies of social existence (Sjöberg et al. 2004). In addition, Dimov and Shepherd (2005) found that risk perception is influenced by the ability to accumulate new knowledge which, in turn, depends on the existing stock of knowledge.

The education sector, as a knowledge-based sector, could be exposed to a variety of knowledge risks. With reference to the Italian school system, one can note that several conditions may expose it to knowledge risks. Excessive bureaucracy (Headley et al. 2021; Kean et al. 2018), for example, could lead to knowledge obsolescence and inefficient teachers' recruitment, and the consequent precariousness (Robillard 2021; Liaropoulos 2020) could involve a progressive loss of valuable knowledge created in the context of ongoing employment relationships. Moreover, Italian high schools are increasingly concerned with complex and burdensome issues, such as the inclusion of students with special educational needs or the fight against early school leaving. Furthermore, the role of teachers has undergone several changes that highlight the need for cultural, psycho-pedagogical, technical, professional, methodological, didactic and relational skills, all in a single professional figure (Beames et al. 2021; Shurygin et al. 2022). Although part of the literature claims the importance of knowledge management within educational institutions (Cheng 2021; Hannum 2001; de Moraes Cordeiro et al. 2022; Raudeliūnienė et al. 2020), there has been little investigation regarding knowledge risks and their perception in the education sector. This study sought to fill this gap, proposing an analysis of the relationship between the demographic characteristics of teachers (TDCs) and their knowledge risk perception (KRP). Furthermore, training as a mediator of the TDCs/KRP relationship was considered as well, as training, the transfer of new knowledge, could indirectly affect risk perception which, in turn, is influenced by the acquisition of new knowledge (Dimov and Shepherd 2005). Therefore, gathering data from a sample of high school teachers in Italy, a research model was developed to relate variables, and the hypothesized relationships were analyzed.

Both theoretical and practical implications can be derived from this paper. Firstly, it adds to research on the relationship between demographics and risk perception and on knowledge risks strands, Secondly, it encourages educational institutions to design training activities aimed at fostering the development of skills useful for improving knowledge risk perception.

The remainder of the paper proceeds as follows. Section 2 outlines the literature review, hypotheses development and theoretical framework of this study. Section 3 includes the methods and measures, and in Section 4, the results of the analysis are provided. Finally, Section 5 discusses the results and concludes the paper, whilst also outlining suggestions for future research.

## 2. Literature Review and Hypotheses Development

### 2.1. Knowledge Risks: Definitions and Main Typologies

Since Durst et al. (2018b) asked "what do we really know about knowledge risks management?", much progress has been made in the study of knowledge risks and in the search for possible solutions to prevent them and mitigate their effects on organizations of all types and sizes. Knowledge risks have been defined as "a measure of the probability and severity of adverse effects of any activities engaging or related somehow to knowledge that can affect the functioning of an organization on any level" (Durst and Zieba 2019, p. 2). Furthermore, knowledge risks have been mapped and classified according to their origin in the human, technological and operational (Durst and Zieba 2019). Human knowledge risks are those originating from human behaviors and, thus, influenced by social, cultural and psychological factors. Technological knowledge risks arise when technological knowledge is not correctly managed, from the employment of obsolete technologies or from the illicit use of technology by cyber criminals. Operational knowledge risks could originate from a variety of firms' operations, such as organizational changes or outsourcing operations (Durst and Zieba 2019). Knowledge loss, knowledge waste, knowledge hiding, knowledge hoarding, forgetting and unlearning are just some of the human knowledge risks currently known (Durst and Zieba 2019). For instance, the risk of knowledge loss may occur in organizations as a result of employee turnover or retirement (Bratianu and Leon 2015; Calo 2008; Massingham 2018; Sumbal et al. 2018; Urbancová and Linhartová 2011). Several authors have proposed methodologies for knowledge loss measurement, the identification

of its determinants and for verifying possible impacts on organizational performance (Eckardt et al. 2014; Jennex 2014; Massingham 2018; Massingham 2008). What makes people hide knowledge has been analyzed in the literature as well (Anand and Hassan 2019; Di Vaio et al. 2021; Issac and Baral 2018). The possible antecedents, consequences and costs of knowledge hiding in organizations have been analyzed (Chatterjee et al. 2021; Khoreva and Wechtler 2020; Xiong et al. 2021), such as the effects of different variables on knowledge hiding behaviors (Arain et al. 2019; Farooq and Sultana 2021; Koay et al. 2022; Nadeem et al. 2020). The risks related to cybercrime, those resulting from the use of old technologies or from digitization and those arising from the improper use of social media have been assigned to the category of technological knowledge risks (Durst and Zieba 2019). Due to the COVID-19 pandemic, organizations have been more frequently exposed to technological knowledge risks, because the outbreak increased the number of people working from home. Not all workers are able to handle technology safely from home, and they can often run risks associated with the lack or improper use of technological knowledge, such as the use of less secure home internet networks, which could expose organizations to cyberattacks (Zięba et al. 2021a). Furthermore, possible inefficiencies in the management of technologies during remote work could affect the work–life balance of employees, as they may need to extend working hours by subtracting time from their private life due to the presence of complications related to the use of technology (Borgia et al. 2022). Operational knowledge risks include knowledge waste risk, risks of knowledge outsourcing, risk of using obsolete/unreliable knowledge and risks resulting from mergers and acquisitions (Durst and Zieba 2019). As a consequence of organizational changes, for example, operational knowledge risks could arise that were not present in the original organizational structure, such as possible vulnerabilities linked to the reorganization of a more complex knowledge management (Borgia and La Torre 2021).

*2.2. Knowledge Risks in Different Operational Contexts*

Knowledge risks have been studied with respect to different perspectives and organizational contexts. Bratianu (2018) expanded the framework for the definition and measurement of knowledge risks by including, in addition to explicit and tacit knowledge, the perspective of rational, emotional and spiritual knowledge, thus arriving at a holistic approach to knowledge risks; while in another study, knowledge vulnerabilities capable of generating knowledge risks were considered as well (Bratianu and Bejinaru 2022). Knowledge risks were also analyzed from the perspective of private (Durst and Henschel 2020) and public (Durst et al. 2018a) companies, SMEs (Durst and Ferenhof 2014), healthcare organizations (Hammoda and Durst 2022) and companies in the financial sector (Durst 2013; La Torre 2020a; Sarigianni et al. 2015; Shujahat et al. 2020). The possible effects of knowledge risks on firms' performance (Durst et al. 2019), firms' commitment to sustainability (Bratianu et al. 2020; Durst and Zieba 2020) and on the effectiveness of staff training (Borgia and La Torre 2021) were addressed as well.

The present review highlights significant research progress on knowledge risks, despite still not being a fully developed strand (Durst 2019). This paper seeks to contribute to the development of this knowledge risk strand, proposing an analysis of the relationship between TDCs and their KRP, considering teachers' training as a mediation variable.

*2.3. Demographics, Risk Perception and Training*

Risk perception relates to the opinions that individuals express when they are asked to define and evaluate dangerous activities and/or situations (Slovic et al. 1982). Considered a valid precursor of the risk reduction process (Bubeck et al. 2012), risk perception concerns the perceived severity of a threat and its perceived probability, and it is positively correlated with previous experiences of danger or with cognitive risk measurements of the probability or severity of a dangerous event (Becker et al. 2014). Risk perception is socially constructed in the sense that it is affected by personal experiences, memory, social status, education level, culture and other subjective factors that influence the way individuals perceive risks,

regardless of the likelihood that risky events occur (Botterill and Mazur 2004). Objective measures are usually employed in risk analyses, while risk perception measurement is entrusted to subjective individuals' assessments (Eryılmaz Türkkan and Hırca 2021). The impact of sociodemographic characteristics on risk perception has been widely studied in the literature. In a study (Gustafsod 1998) in the main literature, the gender and risk perception relationship was reviewed, with the aim of identifying what gender differences were found in such studies and how these differences were accounted. Brown et al. (2021) focused on the implications of gender effects on risk perception at the level of national risk assessment processes of EU members. In other studies (Rana et al. 2021; Rodriguez-Besteiro et al. 2021), gender differences in the COVID-19 pandemic risk perception were analyzed, while in Rosi et al. (2021) and in Kowalsky et al. (2021), COVID-19 risk perceptions were related to age, considering the differences between youths and adults. Age was also considered as a variable of influence on disaster risk perception (Rahman 2019; Seyedin et al. 2019). Eryılmaz Türkkan and Hırca (2021) examined the effect of sociodemographic characteristics on flood risk perception, finding a positive relationship between education and income levels and flood risk perception. Mitchell (1995) discussed the way managers view risk and the factors that affect its perception. The results of another study (Savage 1993) showed that women, young people and people with low income and educational attainment were more fearful of risk and had a greater perception of risk exposure. Using a sample of almost 500 people, Hakes and Viscusi (2004) found that more educated people had a more accurate knowledge of mortality risk, highlighting important differences in risk perception by race and gender. Moreover, demographic determinants of accident experience and related risk perception were identified, finding that men and highly educated respondents perceived their risks to be lower than what was expected considering their incident experience (Sund et al. 2017). Considering the above, the following hypotheses were formulated:

**Hypothesis 1 (H1a).** *Gender affects knowledge risk perception.*

**Hypothesis 1 (H1b).** *Age affects knowledge risk perception.*

**Hypothesis 1 (H1c).** *Education affects knowledge risk perception.*

**Hypothesis 1 (H1d).** *Experience affects knowledge risk perception.*

**Hypothesis 1 (H1e).** *Employment position affects knowledge risk perception.*

Training, together with education, skills, experience, health and personal attributes, was considered as one of the human capital dimensions (Alnoor 2020). Properly selected trained human resources can improve job performance and the performance of an entire organization (Huang et al. 2020; Khan and Quaddus 2018) in addition to improving risk management already connected to perception (Massingham 2010). Zuluaga et al. (2016) found a strong statistical significance between training quality, hazard recognition performance and risk perception, highlighting that efficient training methods can help improve hazard recognition and risk perception. Another study (Rosenbloom et al. 2008) analyzed the effects of advanced driving training on the perception of driving risk. Moreover, knowledge transfer methodologies were considered that allowed a better recognition and perception of the risks and hazards among electrical (Haluik 2016) and construction (Albert and Hallowell 2012; Albert et al. 2020) workers. Taking this into consideration, the following hypotheses were drawn:

**Hypothesis 2 (H2a).** *Training mediates the relationship between gender and knowledge risk perception.*

**Hypothesis 2 (H2b).** *Training mediates the relationship between age and knowledge risk perception.*

**Hypothesis 2 (H2c).** *Training mediates the relationship between education and knowledge risk perception.*

**Hypothesis 2 (H2d).** *Training mediates the relationship between experience and knowledge risk perception.*

**Hypothesis 2 (H2e).** *Training mediates the relationship between employment position and knowledge risk perception.*

The conceptual framework below (Figure 1) schematically represents the hypotheses formulated in this study. According to the framework, TDCs (i.e., gender, age, education, experience and employment position) are the independent variables, KRP is the dependent variable and training is the mediator.

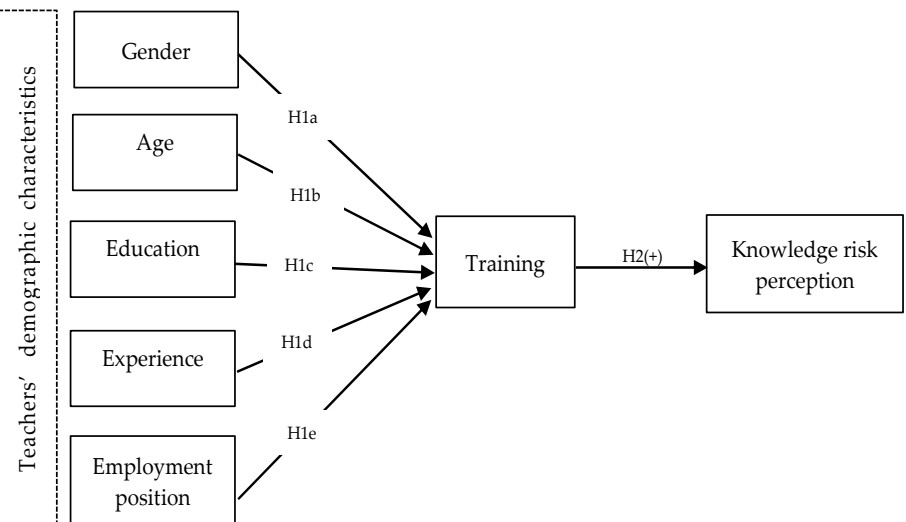

**Figure 1.** The conceptual framework. Source: Authors' conceptualization.

*2.4. Theoretical Framework*

The theories underpinning this study were the protection motivation theory (Rogers 1975) and the human capital theory (Becker 2009). The principle of the protection motivation theory in this study is that individuals, in a risky decision, carry out three cognitive processes, namely, an assessment of the intensity of the threat, a consideration of the probability of its occurrence and an estimation of the ability to face the threat (Le and Arcodia 2018). On the other hand, the proposition of the human capital theory is that the gains of training represent a kind of investment in human resources, i.e., effective HR training is aimed at creating new knowledge which, in turn, could affect risk perception (Nafukho et al. 2004).

**3. Research Methods**

*3.1. Sampling and Data Collection*

Based on the proposed research model, a structured questionnaire was developed (Fife-Schaw 1995) to gather data from teachers in high schools in Italy. The English version of the questionnaire was translated into Italian before it was distributed to the respondents. The questionnaire focused on two main themes: (1) TDCs, namely, age, gender, education, experience, employment position and teachers' specialization; (2) constructs, i.e., KRP and training. Through a specific link administered by email, the online survey was conducted in June 2022 with Italian high school teachers in the region of Abruzzo in Italy. The participants were informed of the purposes of the survey, that anonymity was guaranteed and that the collected data would be used for research purposes only. The sample size was determined using a simple random sampling, and the sampling ratio was fixed to 10%

of the population. One hundred and forty-six respondents were considered suitable for the analysis.

### 3.2. Measures

According to Le and Arcodia (2018), KRP was measured using 3 items adapted from Cunningham's (1967) two-component risk uncertainty and adverse consequences model. Some sample items were "For each knowledge risk presented below, please report your perception of the likelihood that they may occur within your school" and "Which of the following factors and to what extent does it affect knowledge loss in your school organization?". Training was measured using 2 items adapted from Bai et al. (2018) and Jasimuddin et al. (2019); a sample item was "With reference to the learning methods within the school organization (other than the specific curricular knowledge of the teacher), evaluate the effectiveness of the training activities".

### 3.3. Data Analysis Technique

After basic preliminary assumptions, the data analysis was conducted using two-step approaches. Firstly, the degree of association between the TDCs and their KRP was verified using Pearson's chi-squared test (Greenwood and Nikulin 1996). Afterwards, the mediating effect of training on the relationship between TDC and KRP was analyzed through a structural equation model (Hayes 2017). The statistical software STATA was used for the statistical analysis.

## 4. Research Findings

In this section, the findings regarding the demographic characteristics of the respondents and the formulated hypotheses are presented. Table 1 shows the results of the demographics of the participants. From the demographic profiles, it emerged that the respondents were 28.08% men and 71.92% women; 34.25% were between the ages of 51 and 60 and 13.70% were over 60. In addition, 59.59% of the respondents worked as a head teacher, 18.49% worked as a tenured teacher and 11.64% were tenured teachers for disabled people. Regarding the education of the respondents, 3.00% of the respondents were high school graduates, 4.00% had a bachelor's degree and 85.00% had a post-graduate degree. Furthermore, it was found that 39.3% of the respondents had a specialization in a discipline, 28.08% specialized in disabled peoples' needs and 36.99% were not specialized.

**Table 1.** Profiles of the respondents.

| Characteristics | Categories | Frequencies | % |
|---|---|---|---|
| Gender | Male | 41 | 28.08 |
| | Female | 105 | 71.92 |
| Age | 20–30 | 2 | 1.37 |
| | 31–40 | 19 | 13.01 |
| | 41–50 | 55 | 37.67 |
| | 51–60 | 50 | 34.25 |
| | Older than 60 | 20 | 13.70 |
| Education | High school degree | 5 | 3.00 |
| | Bachelor's degree | 6 | 4.00 |
| | Post-graduate degree | 124 | 85.00 |
| | PhD | 11 | 8.00 |
| Teachers' specialization | Specialization in a discipline | 51 | 39.93 |
| | Specialization for disabled people | 41 | 28.08 |
| | No specialization | 54 | 36.99 |
| Employment position | Head teacher | 87 | 59.59 |
| | Tenured teacher | 27 | 18.49 |
| | Tenured teacher for disabled people | 17 | 11.64 |
| | Tenured teacher with fixed term | 15 | 10.27 |

Table 2 shows the main descriptive statistics of the items investigated. In particular, the mean, standard deviation, minimum and maximum were reported. Knowledge, as the values, methodologies and know-how acquired by a school organization (other than curricular-disciplinary knowledge), was reported as more important for 74.65% of the respondents, whilst only 8.21% reported these factors as not important. A total of 55.47% of the teachers interviewed declared that they had never heard about knowledge risk management in their organization, but that wasting, unlearning and hoarding of knowledge were considered the most likely knowledge risks that could be occurring in their school. Moreover, the high level of tenured teachers' mobility was considered the principal factor responsible for knowledge loss in their schools. Teachers' training was recognized as the most effective intervention against knowledge risks followed by other knowledge management tools (self-assessment report, training offer plan, individualized education plans and personalized learning plans). Normality and multicollinearity were essential for the data collected; data were found to meet the standards of normality, and no issue with multicollinearity was found.

**Table 2.** Descriptive statistics.

| Variable | Obs. | Mean | SD | Minimum | Maximum |
|---|---|---|---|---|---|
| Training | 146 | 2.0753 | 0.910367 | 1 | 4 |
| Self-assessment report (SAR) | 146 | 2.1233 | 0.88583 | 1 | 4 |
| Training offer plan (TOP) | 146 | 2.1644 | 0.743036 | 1 | 4 |
| Individualized education plans (IEPs) | 146 | 2.4315 | 0.663403 | 2 | 4 |
| Personalized learning plans (PLPs) | 146 | 2.39726 | 0.689622 | 1 | 4 |
| KRP1 | 146 | 1.4247 | 0.768659 | 1 | 3 |

In Table 3 below, the results of the hypotheses' analysis are presented. The Pearson chi-squared results showed a significative association between gender/KRP, education/KRP and employment position/KRP, while the age/KRP and experience/KRP relationships were found to not be statistically significant. On this basis, a structural equation modeling analysis was conducted to investigate the relationship between TDCs and their KRP, mediated by training. The statistical model had an acceptable fit index ($\chi$2/df = 12.99 (*p*-value = 0.0047), CFI = 0.90, TLI = 0.89 and RMSEA = 0.037). The structural relationships are displayed in Figure 2.

**Table 3.** Chi-squared test results.

| | Pearson Chi2 | *p*-Value |
|---|---|---|
| Gender vs. KRP | 7.991 | 0.018 |
| Age vs. KRP | 7.619 | 0.471 |
| Education vs. KRP | 20.537 | 0.008 |
| Employment position vs. KRP | 12.84 | 0.012 |
| Experience vs. KRP | 4.61 | 0.595 |

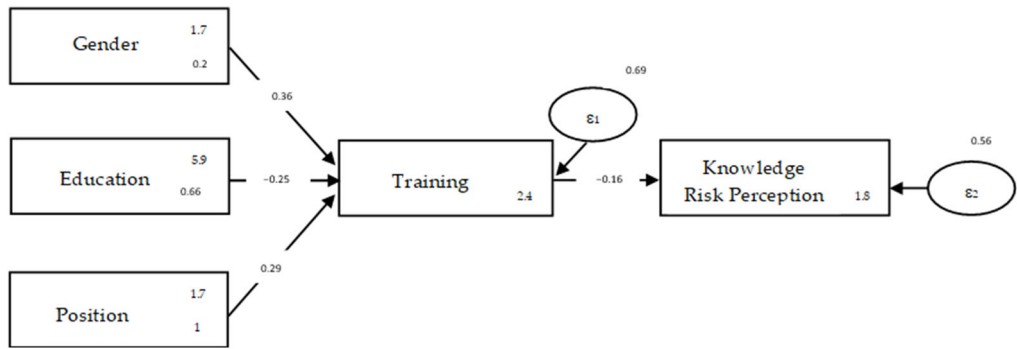

**Figure 2.** Structural equation model. Source: Authors' elaboration.

According to the results of the SEM analysis, gender, education and employment position significantly predicted teachers' KRP. Moreover, one of the most important results was that training negatively predicted almost all KRP (Table 4).

**Table 4.** Estimation structural equation model.

|  | Coefficient | SE | Z | P > |Z| | 95% Confidence Interval | |
|---|---|---|---|---|---|---|
| **Training** | | | | | | |
| Gender | 0.3647982 | 0.158294 | 2.3 | 0.021 | 0.0545486 | 0.675048 |
| Education | 0.2474136 | 0.087550 | −2.38 | 0.005 | −0.4190092 | −0.07582 |
| Employment position | 0.2895783 | 0.067126 | 4.31 | 0.000 | 0.1580134 | 0.421143 |
| Constant | 2.41421 | 0.542709 | 4.45 | 0.000 | 1.35052 | 3.477899 |
| **KRP** | | | | | | |
| Training | −0.1636934 | 0.068552 | −2.39 | 0.017 | −0.2980533 | −0.02933 |
| Constant | 1.764377 | 0.15527 | 11.36 | 0 | 1.460055 | 2.0687 |
| Var(Training) | 0.6863966 | 0.080337 | | | 0.5456945 | 0.865578 |
| Var(KRP) | 0.5647341 | 0.066097 | | | 0.4489771 | 0.710346 |

## 5. Discussion and Conclusions

The findings of this study revealed that demographics had significant effects on teachers' KRP, and that teachers' training mediated this relationship. More specifically, the results of hypothesis one showed that gender, education and employment position were the TDCs with the greatest impact on teachers' KRP. This result is in line with empirical studies that found gender (Brown et al. 2021; Rana et al. 2021; Rodriguez-Besteiro et al. 2021), education (Hakes and Viscusi 2004) and employment position (Mitchell 1995) to have an influence on risk perception. The concordance of these results confirms that risk perception is socially constructed and, thus, affected by personal experiences, memories, social status, education level, culture and other subjective features, regardless of the likelihood that risky events occur (Botterill and Mazur 2004).

The findings of the second hypothesis indicated that teacher training played a mediating role in the relationships between TDCs and KRP. This result is corroborated by the literature that found a strong statistical significance between training and risk perception, regardless of the demographic characteristics and the type of risk (Albert and Hallowell 2012; Albert et al. 2020; Haluik 2016; Rosenbloom et al. 2008). The results of the second hypothesis verify the fact that risk perception does not only concern the personal sphere of individuals but is also influenced by variables such as training and, therefore, could become a valid tool against risks. This is also the case with knowledge risks, which are particularly dangerous for organizations of all types and sizes.

This study, for the first time, addresses the issue of knowledge risks in the context of the secondary education sector. In particular, it aimed to verify the impact of demographics on knowledge risk perception among high school teachers in Italy, considering training as a mediating variable. The findings showed a significant effect of TDCs on KRP, in particular gender, education and employment position, and verified the mediating role of teacher training in the TDCs and KRP relationship. These results contribute to the still underdeveloped knowledge risk management body of knowledge, considering the context of the education sector has not yet been widely investigated in knowledge risks research. Furthermore, this study contributes to the research on the relationship between demographics and knowledge perception by confirming the influence of TDCs on KRP among high school teachers.

This study could also have practical implications, as educational institutions could enhance teacher training activities by paying greater attention to the possible risks deriving from knowledge management, given that teaching is a knowledge-based profession, thus potentially exposed to these risks.

From a macro perspective, knowledge risk management practices could be applied to teachers' recruitment policies to prevent knowledge loss caused by excessive turnover or the precariousness that often characterizes the education sector.

This study's main limitation is the geographical scope, as the high schools are from a single Italian region. Further studies could analyze the knowledge risk perception of teachers from other regions of Italy, and also from other countries, so as to be able to arrive at results that can be generalized. Regarding suggestions for future research, this study suggests that additional variables impacting teachers' knowledge risk perception could be considered and that additional mediating but also moderating variables could be analyzed as well. Furthermore, future research could also concern the possibility of analyzing qualitative data through an ethnographic procedure that more closely explores the experiential dimension of knowledge risks in educational contexts.

**Author Contributions:** Conceptualization, M.B., E.N., M.L.T. and G.O.; methodology, M.B., E.N., M.L.T. and G.O.; software, E.N.; validation, M.B., E.N., M.L.T. and G.O.; formal analysis M.B., E.N., M.L.T. and G.O.; investigation, M.B., E.N., M.L.T. and G.O.; data curation, M.B., E.N., M.L.T. and G.O.; writing—original draft preparation, M.B., E.N., M.L.T. and G.O.; writing—review and editing, M.B., E.N., M.L.T. and G.O.; visualization, M.B., E.N., M.L.T. and G.O.; supervision, M.B., E.N., M.L.T. and G.O. All authors have read and agreed to the published version of the manuscript.

**Funding:** This research received no external funding.

**Institutional Review Board Statement:** Not applicable.

**Informed Consent Statement:** Not applicable.

**Data Availability Statement:** Not applicable.

**Conflicts of Interest:** The authors declare no conflict of interest.

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
