# Peer review of "The Relationship between Demographics and Knowledge Risk Perception of High School Teachers: Training as a Mediator"

_admsci, doi:10.3390/admsci12040188_

Round 1
Reviewer 1 Report
1.It is rather strange that the variable "experience" used in Figure 1 is considered statistically non-significant, because experience contributes through its tacit knowledge in a considerable way to knowledge risk perception. The authors should explain how this result of non-significantly statistic experience could be. In Figure 2, education is "measured" as a result of explicit knowledge development, but "experience" could have contributed with its tacit knowledge. Perception is based on subjective evaluations and tacit knowledge is the key pillar there, not explicit knowledge.
2.Many authors demonstrated that knowledge loss is one of the most important knowledge risk in any organizational management. One of the very efficient methods of reducing knowledge loss and increasing knowledge retention is intergenerational learning through knowledge transfer. We recommend the authors to read: Bratianu, C. & Leon, R.D. (2015). Strategies to enhance intergenerational learning and reducing knowledge loss: an empirical study of universities. VINE Journal of Information and Knowledge Management Systems, 45(4), 551-567. DOI: 10.1108/VINE-01-2015-0007.
3.Using the concept of "demographics" in the title of the paper leads to a fuzzy description of the essence of the paper. It would be more adequate the use of the concept of "human capital" because the characteristics shown in Figure 1 can be better associated with "human capital" than with "demographics". Moreover, "human capital" represents a potential of an organization while "demographics" refers to population not to a given organization.
Reviewer 2 Report
The article provides an abundant, varied and updated bibliography, which justifies the originality that this topic represents in the field of educational research. Figure 1 is very helpful in understanding the approach of the study and the design of the questionnaire used is well justified.
However, it is essential to improve the text organisation by dividing some very long paragraphs (e.g. at the beginning of sections 1 and 2.1) into several shorter paragraphs. This will allow the reader to better follow the main ideas.
The end of the introduction section suggests some ideas that should be moved to the methodology section or could be deleted, as they seem to be, in a way, a summary of the results.
References in the same bracket should be ordered correctly, in alphabetical order.
"et al." should be used whenever there are more than two authors and not only sometimes.
Authors should delete the word "the" at the beginning of the discussion.
Considering the short length of the discussion (despite the length of the first theoretical sections), the recommendation is to expand this section a little more and even combine it with the conclusions ("Discussion and conclusions"). In fact, some of the ideas that appear in the current conclusions should be discussed as well.
Finally, regarding the limitations of the research, I suggest including some reflection on the possibility of analysing qualitative data through an ethnographic procedure that explores more closely the experiential dimension of knowledge risks in educational contexts.
